# Effect of Equivalence Ratio on Pollutant Formation in CH_4_O/H_2_/NH_3_ Blend Combustion

**DOI:** 10.3390/molecules29010176

**Published:** 2023-12-28

**Authors:** Jingyun Sun, Qianqian Liu, Mingyan Gu, Yang Wang

**Affiliations:** 1School of Energy and Environment, Anhui University of Technology, Ma’anshan 243002, China; jingyunsun133@163.com (J.S.); liuqianqian208dw@163.com (Q.L.); gumy@ahut.edu.cn (M.G.); 2School of Materials Science and Engineering, Anhui University of Technology, Ma’anshan 243032, China

**Keywords:** ternary blend combustion, equivalent ratio, NO_X_, ReaxFF MD, reaction mechanism

## Abstract

This paper investigates the effect of equivalence ratio on pollutant formation characteristics of CH_4_O/H_2_/NH_3_ ternary fuel combustion and analyzes the pollutant formation mechanisms of CO, CO_2_, and NO_X_ at the molecular level. It was found that lowering the equivalence ratio accelerates the decomposition of CH_4_O, H_2_, and NH_3_ in general. The fastest rate of consumption of each fuel was found at φ = 0.33, while the rates of CH_4_O and NH_3_ decomposition were similar for the φ = 0.66 and φ = 0.4. CO shows an inverted U-shaped trend with time, and peaks at φ = 0.5. The rate and amount of CO_2_ formation are inversely proportional to the equivalence ratio. The effect of equivalence ratio on CO_2_ is obvious when φ > 0.5. NO_2_ is the main component of NO_X_. When φ < 0.66, NO_X_ shows a continuous increasing trend, while when φ ≥ 0.66, NO_X_ shows an increasing and then stabilizing trend. Reaction path analysis showed that intermediates such as CH_3_ and CH_4_ were added to the CH_4_O to CH_2_O conversion stage as the equivalence ratio decreased with φ ≥ 0.5. New pathways, CH_4_O→CH_3_→CH_2_O and CH_4_O→CH_3_→CH_4_→CH_2_O, were added. At φ ≤ 0.5, new intermediates CHO_2_ and CH_2_O_2_ were added to the CH_2_O to CO_2_ conversion stage, and new pathways are added: CH_2_O→CO→CHO_2_→CO_2_, CH_2_O→CO→CO_2_, CH_2_O→CHO→CO→CHO_2_→CO_2_, and CH_2_O→CH_2_O_2_→CO_2_. The reduction in the number of radical reactions required for the conversion of NH_3_ to NO from five to two directly contributes to the large amount of NO_X_ formation. Equivalent ratios from 1 to 0.33 corresponded to 12%, 21.4%, 34%, 46.95%, and 48.86% of NO_2_ remaining, respectively. This is due to the fact that as the equivalence ratio decreases, more O_2_ collides to form OH and some of the O_2_ is directly involved in the reaction forming NO_2_.

## 1. Introduction

Currently, the global transportation industry relies mainly on fossil energy sources [1], but the combustion of these traditional fossil energy sources causes a lot of pollution. Clean, efficient, and sustainable are the current trends in energy development [2,3]. Hydrogen and ammonia are both ideal clean and renewable fuels, which have received extensive attention from scholars at home and abroad. Hydrogen is renewable and characterized by good combustibility, low ignition energy, and fast combustion speed [4,5]. However, the difficulties in storing and transporting hydrogen, the premature ignition and backfire caused by overly fast combustion speeds, and the high combustion temperature that produces NO_X_ pollution have all limited the practical popularization of the use of pure hydrogen fuel [6]. Ammonia, as a good zero-carbon hydrogen storage carrier, can be obtained from biomass or other renewable sources. It is considered a sustainable fuel that can be transported and applied remotely [7]. Currently, ammonia is widely used as a fuel in automobile engines [8], marine engines [9], and generator internal combustion engines [10], where the low viscosity of ammonia helps in fuel atomization and droplet formation during fuel injection [11]. Ammonia also has a high octane rating, which makes it suitable for engines with high compression ratios and reduced detonation [12]. However, the disadvantages of ammonia’s low combustion rate [13] and high auto-ignition temperature [14], as well as narrow combustible limits, tend to lead to incomplete combustion, which results in poor engine performance. Therefore, it is difficult to use as a single fuel for direct combustion [15,16]. The use of hydrogen as a combustion aid and ammonia miscombustion was found to be one of the ways to improve ammonia combustion efficiency [17]. This not only leads to improved in-cylinder combustion [18] but also reduces the requirement for engine modifications (material compatibility), thus ensuring a cost-effective transition to hydrogen energy [19]. Wang et al. [20] found that engine exhaust heat can crack some of the ammonia into hydrogen and nitrogen to form reformed gases, making this method much more maneuverable. A study by Alam et al. [21] indicated that although hydrogen–ammonia blending can reduce carbon emissions including CO and others in diesel internal combustion engines, incomplete combustion of the fuel and higher NO_X_ emission phenomena were observed.

Blending oxygenated fuels as a combustion aid is also an effective way to improve combustion performance and pollutant emissions in diesel engines [22,23]. Methanol, as the saturated monohydric alcohol with the simplest structure, is inexpensive and simple to synthesize. It is a high-quality representative for studying the combustion-enhancing effect of oxygenated fuels [24,25]. Methanol is ideal for fuel-lean combustion. However, obtaining high energy and reliable ignition is one of the biggest challenges of fuel-lean combustion [26]. The reformed gas in the engine can provide exactly this energy due to the presence of H_2_. Li et al. [27] investigated the ignition delay time of ammonia/methanol blends with equivalence ratios of 0.5, 1.0, and 2.0 and temperatures in the temperature range of 1250–2150 K. The results showed that the ignition delay time of ammonia/methanol blends was mainly affected by free radicals such as OH, O, HO_2_, and H. Li et al. [28] found that blending a small amount of methanol into ammonia combustion made the blend more reactive due to the fact that the addition of methanol introduced a new reaction sequence, CH_3_OH→CH_2_OH/CH_3_O→CH_2_O→CHO, which enriched the O/H radical library.

However, there are very few studies on CH_4_O/H_2_/NH_3_ blend combustion. Given the complexity of engine in-cylinder combustion and pollutant formation characteristics, it is not conducive to the isolated exploration of chemical reaction kinetics and mixed fuel combustion pollutant laws under different operating parameters [29]. This leads to the fact that the mechanism of blended combustion action is not yet well clarified.

## 2. Results and Discussion

### 2.1. Effect of Equivalent Ratio on Combustion Components of Ternary Carbon-Neutral Fuel Blends

Figure 1 shows the effect of different equivalence ratios on the four reactant components, CH_4_O, NH_3_, H_2_, and O_2_, during the blended combustion process of ternary carbon-neutral fuels. Lowering the equivalence ratio accelerates the decomposition of CH_4_O, NH_3_, and H_2_ in general. As the equivalence ratio is lowered, the decomposition rate of CH_4_O is the fastest at φ *=* 0.33 throughout the reaction. As the reaction proceeds, the decomposition rate of the φ *=* 0.5 condition becomes progressively higher, gradually replacing the φ *=* 0.4 condition. At this time, the decomposition rate of CH_4_O was similar between φ *=* 0.4 and φ *=* 0.66. The decomposition rate of NH_3_ increased linearly with the decrease in the equivalence ratio, and the consumption rate of O_2_ increased with the increase in the equivalence ratio when φ ≤ 0.5, and its consumption rate was the smallest when φ *=* 0.4. The curves of H_2_ showed a similar trend to that of CH_4_O, and the highest consumption rate was found in the case of φ *=* 0.33; this was next to that in the case of φ *=* 0.5, but it was different from that of φ *=* 0.4. φ *=* 0.4 is not notably different.

Figure 2 shows the variation of major products and radicals during combustion of ternary carbon-neutral fuels at different equivalence ratios. Figure 2a indicates that there is almost no change in N_2_ with time for different equivalence ratios. Only N_2_ at φ *=* 1 has an increase, and the decreasing trend of N_2_ becomes more and more obvious as the equivalence ratio decreases at φ ≤ 0.66. This is because at a reaction temperature of 2000 K, oxygen becomes more and more abundant as the equivalence ratio decreases, and more N_2_ reacts with O to produce more thermodynamic NO_X_. Figure 2b shows the trend of H_2_O over time. There is no strict linear relationship between the amount of H_2_O generated and the equivalence ratio. The maximum amount of H_2_O is generated at φ *=* 0.66, and there is little difference between φ *=* 1 and φ *=* 0.5.

Figure 2c,d show the effect of different equivalence ratios on the formation of OH and H during the blending process of ternary carbon-neutral fuels, respectively. Comparing the two figures, it can be seen that the effect of equivalence ratio on OH is more pronounced. OH increases rapidly and then decreases slowly as time progresses. The higher the equivalence ratio, the higher the amount of low OH. OH may be the key radical leading to the depletion of CH_4_O, H_2_, and NH_3_. This conclusion will be confirmed in Section 2.3. The H curve shows a tendency to rise and then fall, with a small but fluctuating overall number. The peak occurs at φ = 0.66. H also assumes an important role in the reaction.

### 2.2. Effect of Equivalence Ratio on Pollutant Formation in Blended Combustion of Ternary Carbon-Neutral Fuels

#### 2.2.1. Effect of Equivalent Ratio on CO and CO_2_ Formation in Blended Combustion of Ternary Carbon-Neutral Fuels

Figure 3a shows the formation of CO during the blending process of ternary carbon-neutral fuels at different equivalence ratios. CO shows an inverted U-shape trend with time, the peak value of CO shifts backward with the increase in the equivalence ratio, and the rate of CO formation increases with the decrease in the equivalence ratio in the early stage of the reaction. The CO peaks were 9.33, 8.67, 9.67, 9.33, and 8.67 for the equivalence ratios from 1 to 0.33, respectively. The maximum CO peak was observed at φ = 0.5. This may be due to the fact that there is more CO production and less CO consumption at φ = 0.5. The detailed pathway analysis will be carried out at the molecular level in Section 2.3 for the specific causes.

Figure 3b shows the CO_2_ formation during the combustion of ternary carbon-neutral fuel blends with different equivalence ratios. The CO_2_ formation rate and amount are inversely proportional to the size of the equivalence ratio. The equivalence ratio has little effect on the amount of CO_2_ when φ ≤ 0.5. When the equivalence ratio φ > 0.5, the effect of the equivalence ratio on CO_2_ is more obvious.

#### 2.2.2. Effect of Equivalent Ratio on NO_X_ Formation in Blended Combustion of Ternary Carbon-Neutral Fuels

Figure 4 shows the effect of equivalence ratio on the formation of NO_X_ (NO, NO_2_, and NO_3_) in the combustion of ternary carbon-neutral fuel blends. From Figure 4a, it can be seen that as the combustion proceeds, NO shows a trend of rapid increase followed by a slow decrease. The peak value of NO increases with the decrease in the equivalence ratio. From Figure 4b,c, it can be seen that both NO_2_ and NO_3_ gradually increase with the reaction; NO_2_ is the main component of NO_X_. NO_3_ shows an overall trend of increasing and then slowly decreasing, and the peak value increases with the decrease in the equivalence ratio, and the peak time is also delayed. In the middle and late stages of the reaction, NO_3_ at φ = 0.33 was significantly higher than other working conditions.

As can be seen from Figure 4d, when φ *≥* 0.66, NO_X_ shows a tendency to increase and then stabilize as the reaction proceeds. When φ *<* 0.66, NO_X_ shows a continuous growth trend. and the growth rate decreases around 200ps. However, the NO_X_ growth rate in the middle and late stages when φ *<* 0.44 is significantly higher than that in the case of φ *≥* 0.44.

### 2.3. Mechanism Analysis of CO, CO_2_, and NO_X_ Formation in the Combustion of Ternary Blended Fuel as Affected by Equivalence Ratio

In order to further discuss the impact of ternary blended fuel combustion on the mechanism of CO, CO_2_, and NO_X_ formation as affected by the equivalence ratios, this paper generates reaction network diagrams for five operating conditions and discusses the N and C migration paths of ternary blended fuel combustion at different equivalence ratios as simulated using ReaxFF MD. Figure 5 represents the network diagrams of CO and CO_2_ formation paths during the combustion of ternary carbon-neutral fuels at equivalence ratios of 1, 0.66, 0.5, 0.4, and 0.33, respectively. The percentage in the network diagram indicates the reactant conversion rate in order to highlight the main paths of the reaction network, and the reaction paths with a conversion rate of less than 15% are ignored in all network diagrams in this study.

As can be seen in Figure 5a, all of the CH_4_O is converted to CH_2_O at φ *=* 1. A proportion of 40% of the CH_2_O is generated as CO, and 77% of the CH_2_O is converted to CO_2_. This is consistent with the numerical ratios of CO and CO_2_ in Section 2.2. From Figure 5b, it can be seen that all CH_4_O is also converted to CH_2_O at φ *=* 0.66. The difference with φ *=* 1 is that, in this case, CH_4_O undergoes a direct reduction reaction with H, and this reaction produces CH_3_. The conversion of CH_2_O to CO and CO_2_ in this case is both 20%.

As can be seen from Figure 5c, the complexity of the reaction path at φ = 0.5 is mainly reflected in the transition from CH_4_O to CH_2_O. There are three main paths in this part, which are: CH_4_O→CH_3_O→CH_2_O, CH_4_O→CH_3_→CH_2_O, and CH_4_O→CH_3_→CH_4_→CH_2_O. Among them, CH_3_ and CH_4_ can also be converted to each other. In terms of conversion rate, only 80% of CH_4_O is converted to CH_2_O through intermediates such as CH_3_O, CH_3_, and CH_4_. Statistically, 69% of CH_4_O is converted to CO. A total of 37% of CH_4_O is converted to CO_2_. From Figure 5d, it can be seen that all of the CH_4_O is converted to CH_2_O when φ = 0.4. The conversion rates of CH_2_O to CO and CO_2_ are 60.2% and 46.5%, respectively. From Figure 5e, it can be seen that at φ = 0.33, 80% of CH_4_O is converted to CH_2_O from CH_3_O. The conversion rates of CH_4_O to CO and CO_2_ are 32% and 80%, respectively. The path diagram for this case is also complex, unlike at φ = 0.5, where the complexity is mainly in the conversion phase of CH_2_O to CO and CO_2_. There are four main reaction paths in this stage, namely CH_2_O→CHO_2_→CO_2_, CH_2_O→CO→CO_2_, CH_2_O→CHO→CO→CO_2_, and CH_2_O→CH_2_O_2_→CO_2_.

Comparing with Figure 5, it is found that the number of pre-reaction paths increases as the equivalence ratio decreases for φ ≥ 0.5. At φ = 1, there are only two paths from CH_4_O to CH_2_O, CH_4_O→CH_2_O and CH_4_O→CH_3_O→CH_2_O. At φ = 0.66, the path of direct conversion of CH_4_O to CH_2_O disappears, and the new path CH_4_O→CH_3_→CH_2_O is added. At φ = 0.5, the new path CH_4_O→CH_3_→CH_4_→CH_2_O is added compared with φ = 0.66. Combined with Figure 2d, this is because there is more H at φ = 0.5 and φ = 0.66. For φ ≤ 0.5, the variety of paths in the later stages of the reaction increases as the equivalence ratio decreases. The intermediate CHO_2_ is added at φ = 0.5 compared to φ > 0.5. The reaction paths from CH_2_O to CO_2_ are only CH_2_O→CHO→CO→CO_2_ and CH_2_O→CHO_2_→CO_2_. The new paths CH_2_O→CO→CHO_2_→CO_2_, CH_2_O→CO→CO_2_, and CH_2_O→CHO→CO→CHO_2_→CO_2_ are added at φ = 0.4 compared with φ = 0.5. The new paths CH_2_O→CO→CO_2_ and CH_2_O→CH_2_O_2_→CO_2_ are added at φ = 0.33. Statistics show that the highest CO production rate is achieved at φ = 0.5. This validates the conclusion in Section 2.1 that the peak CO occurs at φ = 0.5. The equivalence ratios from 0.66 to 0.33 correspond to CO_2_ production rates of 20%, 36.9%, 46.5%, and 80%, respectively. The increase with decreasing equivalence ratio is in line with the trend of CO_2_ formation observed in Section 2.2. It was also found that the lowest CO and CO_2_ production rates were both 20% at φ = 0.66, and their consumption rates were also the lowest. The combined analysis reveals that the lowest percentage of total CO and CO_2_ remaining is found at φ = 0.66. Analyzed in conjunction with Figure 2c,d, this is the result of the higher H/OH ratio at φ = 0.66.

Figure 6 represents the network diagram of NO_X_ formation reaction paths in the combustion process of ternary fuels at equivalence ratios of 0.1, 0.66, 0.5, 0.4, and 0.33, respectively. As can be seen from the figure, all NO_X_ in the reaction is converted from NO. As can be seen from Figure 6a, the reaction generates more N_2_ at φ = 1. There are four main paths of N_2_ formation. They are NH_3_→N_2_H_5_→N_2_, NH_3_→NH_2_→N_2_H→N_2_, NH_3_→NH_2_→HNO→N_2_, and NH_3_→NH_2_→NH→N_2_. This is a result of the fact that less OH radicals are generated by the lower O_2_ at the high equivalence ratios. NO→HNO_2_→NO_2_ is the main path in this case. From Figure 6b, 60% of NH_3_ is converted to NO at φ = 0.66. NH_3_→NH_2_→NH→HNO→NO→HNO_2_→NO_2_ is the main conversion path. Compared with φ = 1, a new pathway of NO_3_ formation and consumption is added: NH_3_→NH_2_→NH→HNO→NO→HNO_2_→NO_2_→HNO_3_→NO_3_→NO_2_. From Figure 6c, it can be seen that at φ = 0.5, NH_3_ is fully converted to NO through two different pathways: NH_3_→NH_2_→HNO→NO and NH_3_→NH→HNO→NO. This also leads to the subsequent production of more NO_X_. As can be seen from Figure 6d, the conversion of NH_3_ to NO is reduced to 65% at φ = 0.4. There are also two main paths: NH_3_→NH_2_→HNO→NO and NH_3_→HNO→NO. The path from NH_3_ to NO is shorter compared to that at φ = 0.5. From Figure 6e, the main path is NH_3_→NH_2_→H_2_NO→NO→HNO_2_→NO_2_→NO_3_ at φ = 0.33. The conversion rate of NH_3_ to NO is 80%. The two conversion paths are NH_3_→NH_2_→NO and NH_3_→HNO→NO. Fewer intermediates are required for the conversion of NH_3_ to NO in this case than in other cases, and the conversion of NO to NO_3_ is more direct: NO→NO_2_→NO_3_.

A comparison of Figure 6 shows that the main conversion path of NH3 to NO shifts from NH_3_→NH_2_→NH→NO, NH_3_→NH_2_→NH→HNO→NO, and NH_3_→NH_2_→HNO→NO to NH_3_→HNO→NO and NH_3_→NH_2_→NO as the equivalence ratio decreases. The reaction path becomes progressively shorter, which is caused by more O_2_ in the reaction as the equivalence ratio decreases. With more O_2_, more OH and O radicals are produced in the reaction, and at low equivalence ratios, O_2_ also participates directly in the reaction as a free radical. NO_2_ is the main component of NO_X_. Statistics show that the remaining proportions of NO_2_ corresponding to equivalence ratios from 1 to 0.33 are 12%, 21.4%, 34%, 46.95%, and 48.86%, respectively. The remaining proportion of NO_2_ increases with decreasing equivalence ratios, which explains the conclusion of Section 3.1 that the amount of NO_2_ increases with decreasing equivalence ratios. The main reaction paths for each case are NH_3_→NH_2_→NH→NO→HNO_2_→NO_2_, NH_3_→NH_2_→NH→HNO→NO→HNO_2_→NO_2_→NO, NH_3_→NH_2_→HNO→NO→NO_2_→NO, NH_3_→HNO→NO→HNO_2_→NO_2_→NO, and NH_3_→NH_2_→H_2_NO→NO→HNO_2_→NO_2_→NO_3_. Only φ = 0.33 contains NO3 production in the main pathway. Combined with the conversion analysis, the conversion of NH_3_ to NO_3_ is 0, 2.9%, 8.5%, 0, and 19.95%, respectively. This is due to the fact that there are more OH radicals in the tether at low equivalence ratios. It explains the observation in Section 2.1 that NO_3_ is much higher at φ = 0.33 than other cases.

## 3. Materials and Methods

### 3.1. Reactive Force Field Molecular Dynamics (ReaxFF MD)

ReaxFF MD is a molecular dynamics simulation combined with the calculation of reaction force fields. Its reactive force field potential function is derived from experimental data and density functional theory, so the accuracy is close to quantum computation and does not require the predetermination of chemical reaction paths in the system [30]. ReaxFF MD has been widely used in the study of pyrolysis [31], combustion [32], explosions [33], oxidation [34], catalytic [35], and other systems involving physical chemistry. It provides a promising means of exploring the chemical behavior of complex molecular systems. Bond-order-dependent characterization is achieved through detailed parameterization of the atomic, bonding, angular, and torsional properties of each particle, and the interactions within the system [36]. The total energy of the system can be calculated by summing all partial energy terms as described in R1:E_system_ = E_bond_ + E_over_ + E_under_ + E_val_ + E_pen_ + E_tors_ + E_conj_ + E_vdWaals_ + E_coulomb_(1)
where E_bond_, E_over_, E_under_, E_val_, E_pen_, E_tors_, and E_conj_ correspond to bond energy, over-coordination energy, under-coordination energy, bond angle energy, compensation energy, torsion energy, and four-body conjugation energy. The non-bonding terms mainly consist of van der Waals force energy (E_vdWaals_) and Coulomb force energy (E_coulomb_). When calculating non-bonding interactions, the charged atoms cross the truncation radius of the non-bonding interactions, thus leading to a jump in energy. Therefore, the ReaxFF force field is additionally corrected by introducing a seventh-order polynomial Taper function, which ensures that at the truncation radius, the non-bonding interaction’s first-, second-, and third-order derivatives of the energy term are all zero [37]. The ReaxFF force field also takes better account of charge polarization by employing the electronegativity equalization method [38] and updates the atomic charges at each time step [39]. The detailed meaning of the ReaxFF force field parameters, the setup of the molecular structure, and the applicability of the reaction force field have been described in detail in a previous study [40].

### 3.2. Case Set-Ups

Table 1 lists all the CH_4_O/H_2_/NH_3_ blended combustion ReaxFF MD simulation cases under the high-pressure environment in this paper. The system density (ρ), temperature (T), and simulation time are 0.1 g/cm^3^, 2000 K, and 1.25 ns, respectively. Cases 1 to 5 denote the combustion of CH_4_O/H_2_/NH_3_ blends at fuel equivalent ratios (φ) of 0.5, 1, 0.66, 0.4, and 0.33, respectively. Each condition is calculated three times, keeping the initial settings constant. All results in this paper are averaged over three simulations. Through further comparative analyses, the mechanisms of CO, CO_2_, and NO_X_ pollutant formation at different equivalence ratios are analyzed at the molecular level.

### 3.3. Computational Details and Post-Processing

All the cases listed in Table 1 were carried out in the ReaxFF module of AMS [41,42,43]. In this study, the HE2.ff force field [44] and the regular system with constant atomic number, volume, and temperature (NVT) were used. To ensure the overall stability of hydrocarbon fuel combustion, the energy, and configuration of all simulated cases were first optimized using the “Geometry Optimization” and “Energy Optimization” plug-ins. Figure 7 shows the optimized systematic for case 1, which shows that the fuel and oxidant are uniformly blended, similar to a premixed flame, and similar to the cyclone burner we previously employed [45]. A Berendsen thermostat was used to control the temperature with a time step of 0.25 fs. Periodic boundary conditions were applied in all three xyz directions, and the soot intermediate components and product distributions were analyzed from trajectories using a 0.3 Å bond level cut-off.

### 3.4. Validation of the ReaxFF MD Method

The reliability and validity of the ReaxFF MD method have been widely used and verified in previous studies [36,37,46,47,48,49]. Among them, Wang et al. [36] constructed the reaction pathway of high-pressure combustion by tracking the trajectories of reacting atoms through ReaxFF MD. To understand the NO_X_ formation mechanism of NH_3_/CH_4_ combustion at different temperatures and pressures. The results showed that the high temperature accelerated the rate of NH_3_ consumption, which was consistent with the experimental results. The high pressure complicated the reaction pathway of NH_3_/CH_4_ combustion through the emergence of new intermediates and primitive reactions. In addition, they pointed out that ReaxFF MD is a valuable tool for revealing the reaction mechanisms of combustion and pollutant formation in depth. Liu et al. [49] investigated the chemical reactivity effects of NO on the oxidation of CH_4_ using ReaxFF MD simulations and found that increasing the blending ratio of NO accelerated the rate of CH_4_ consumption. This is mainly due to the fact that, on the one hand, the conversion of NO to NO_2_ generates OH radicals, which accelerates the CH_4_ consumption; on the other hand, NO can also inhibit the CH_4_ consumption by combining with reactive radicals. Wang et al. [46] applied ReaxFF MD and Py-GC/MS to investigate the characteristics of the soot particulate formation in the process of hydrogen-doped combustion of methane and ethylene, and both experimental and numerical results reflected that PAHs and ethylene were not the most important pollutants in the combustion process of CH_4_. The experimental and numerical results reflect the evolution of PAHs and initial soot particles, as well as the different chemical effects of hydrogen doping on PAHs and soot formation.

## 4. Conclusions

In this paper, the effects of different reactant equivalence ratios on the combustion reaction rates and the formation characteristics of CO, CO_2_, and NO_X_ pollutants during the combustion of CH_4_O/H_2_/NH_3_ ternary carbon-neutral blended fuels have been investigated for the first time using ReaxFF MD. The mechanisms of CO, CO_2_, and NO_X_ formation in ternary blended fuels with different equivalence ratios were investigated at the molecular level. The conclusions of this paper are summarized as follows:(1)Reducing the equivalence ratio accelerates the decomposition of CH_4_O, NH_3_, and H_2_ in general. The rate of consumption of each fuel is fastest at φ = 0.33. The rates of CH_4_O and NH_3_ decomposition are similar at φ = 0.66 and φ = 0.4.(2)CO showed an “inverted U” shaped trend of increasing and then decreasing over time. The CO peak appeared at φ = 0.5. CO_2_ shows a continuous increase as the reaction proceeds. The rate and amount of CO_2_ formation are inversely proportional to the magnitude of the equivalence ratio. When φ > 0.5, the effect of equivalence ratio on CO_2_ is more obvious. NO_2_ is the main component of NO_X_. When φ ≥ 0.66, NO_X_ shows a tendency to increase and then stabilize as the reaction proceeds. When φ < 0.66, NO_X_ shows a continuous increasing trend.(3)C migration path analysis showed that for φ ≥ 0.5, the intermediates CH_3_ and CH_4_ are added to the CH_4_O to CH_2_O conversion stage as the equivalence ratio decreases. The new pathways are CH_4_O→CH_3_→CH_2_O and CH_4_O→CH_3_→CH_4_→CH_2_O. At φ ≤ 0.5, new intermediates CHO_2_ and CH_2_O_2_ are added to the CH_2_O to CO_2_ phase as the equivalence ratio decreases. The added paths are CH_2_O→CO→CHO_2_→CO_2_, CH_2_O→CO→CO_2_, CH_2_O→CHO→CO→CHO_2_→CO_2_, and CH_2_O→CH_2_O_2_→CO_2_.(4)N migration pathway analysis showed that the conversion pathway of NH_3_ to NO shifted from the long reaction chains of NH_3_→NH_2_→NH→NO, NH_3_→NH_2_→NH→HNO→NO, and NH_3_→NH_2_→HNO→NO, to the shorter reaction chains of NH_3_→HNO→NO and NH_3_→NH_2_→NO as the equivalence ratio decreased. This is due to the fact that as the equivalence ratio decreases, more O_2_ collides to form OH and some of the O_2_ is directly involved in the reaction. NO_2_ is the main component of NO_X_. Statistics show that the equivalence ratios from 1 to 0.33 correspond to 12%, 21.4%, 34%, 46.95%, and 48.86% of NO_2_ remaining, respectively. This is also caused by the influence of the equivalence ratio on the OH radical concentration.

## Figures and Tables

**Figure 1 molecules-29-00176-f001:**
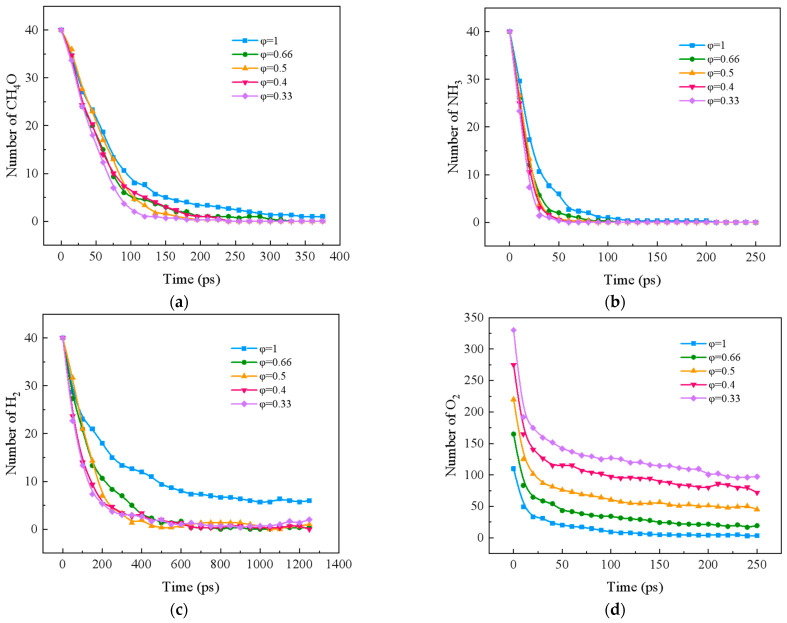
Changes in reactants during combustion of carbon-neutral fuels with different equivalence ratios. (**a**) CH_4_O; (**b**) NH_3_; (**c**) H_2_; (**d**) O_2_.

**Figure 2 molecules-29-00176-f002:**
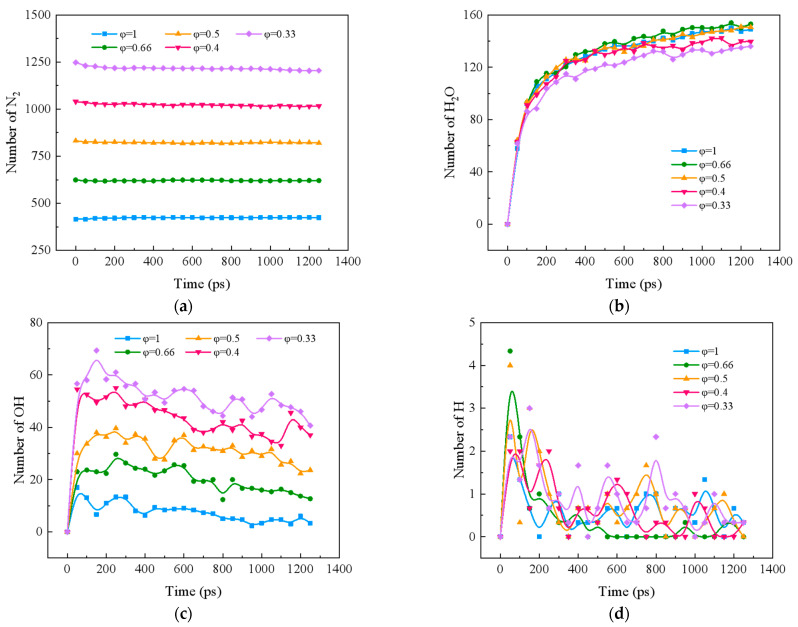
Changes of components and radicals during combustion at different temperatures. (**a**) N_2_; (**b**) H_2_O; (**c**) OH; (**d**) H.

**Figure 3 molecules-29-00176-f003:**
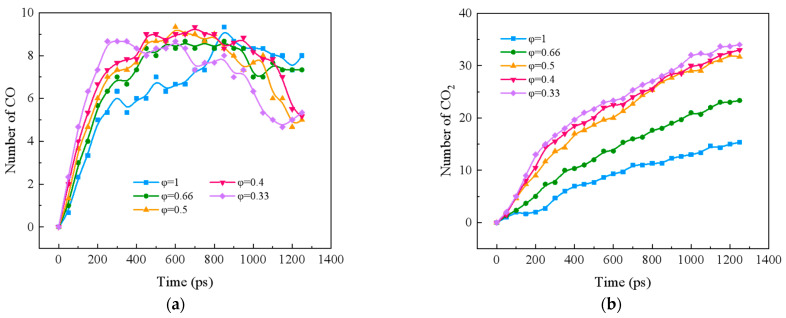
CO and CO_2_ formation with time for blended combustion. (**a**) CO; (**b**) CO_2_.

**Figure 4 molecules-29-00176-f004:**
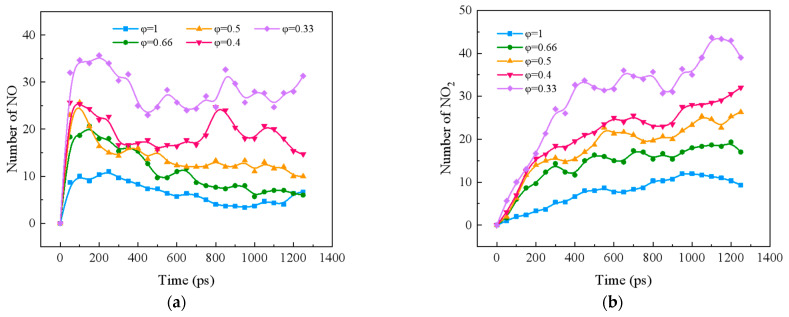
NO_X_ distribution during combustion of ternary carbon-neutral fuel blends with different equivalence ratios. (**a**) NO; (**b**) NO_2_; (**c**) NO_3_; (**d**) NO_X_.

**Figure 5 molecules-29-00176-f005:**
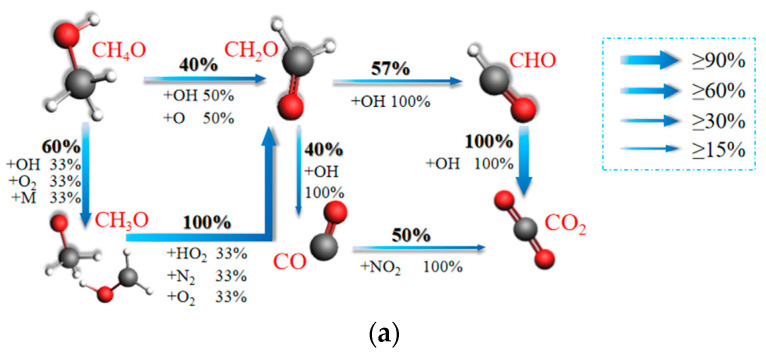
Main migration paths of C in ternary carbon-neutral fuel blends at different equivalence ratios. (**a**) φ = 1; (**b**) φ = 0.66; (**c**) φ = 0.5; (**d**) φ = 0.4; (**e**) φ = 0.33.

**Figure 6 molecules-29-00176-f006:**
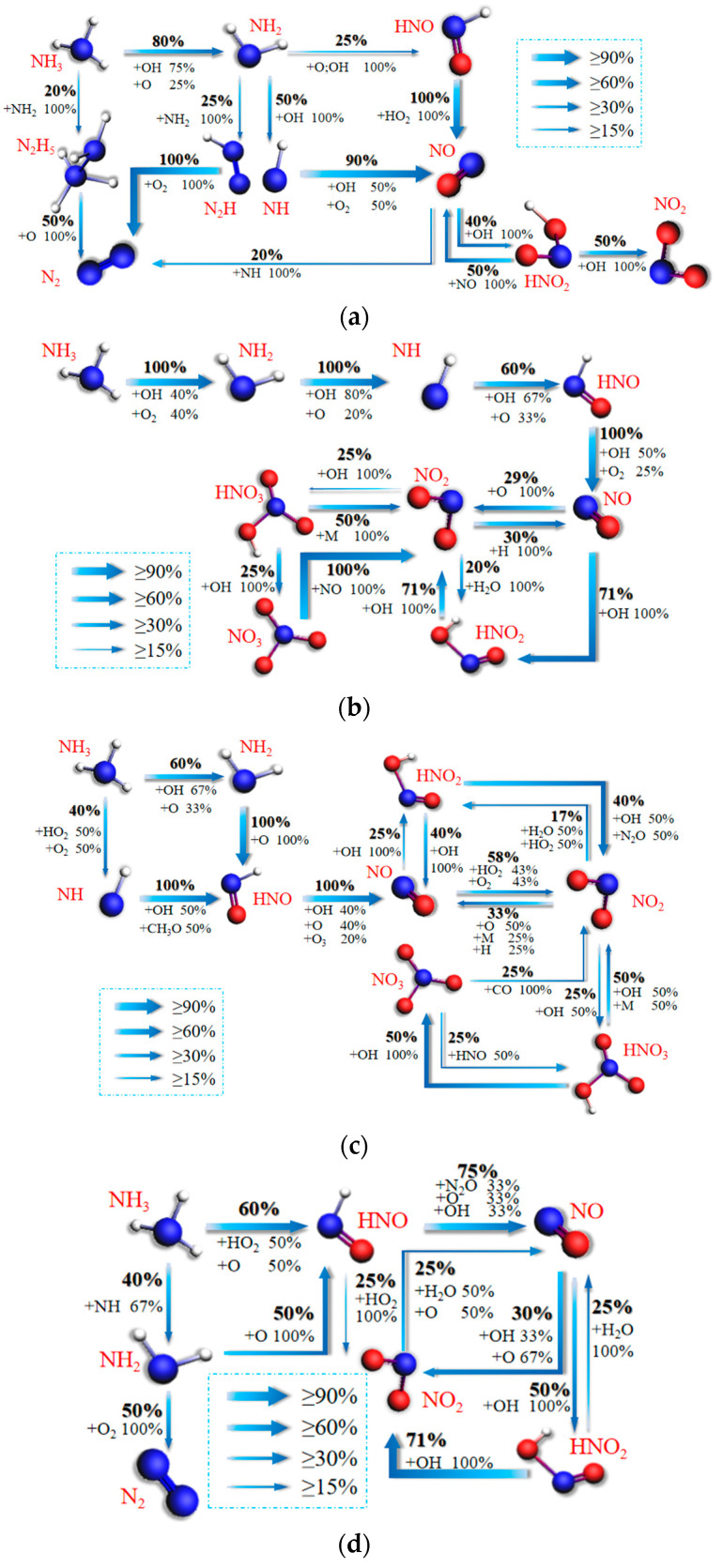
Main migration paths of N in ternary carbon-neutral fuel blends at different equivalence ratios. (**a**) φ = 1; (**b**) φ = 0.66; (**c**) φ = 0.5; (**d**) φ = 0.4; (**e**) φ = 0.33.

**Figure 7 molecules-29-00176-f007:**
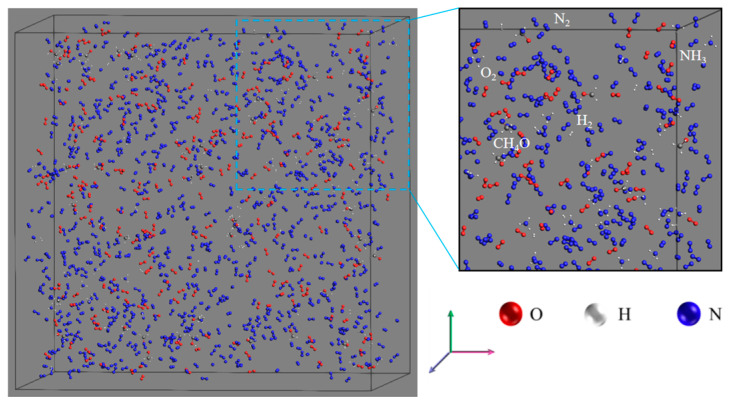
Optimization system for case 1.

**Table 1 molecules-29-00176-t001:** ReaxFF MD cases of the CH_4_O/H_2_/NH_3_ blended combustion.

Case	CH_4_O	H_2_	NH_3_	O_2_	N_2_	ρ, g/cm^3^	T, K	Φ
1	40	40	40	220	832	0.1	2000	0.5
2	40	40	40	110	416	0.1	2000	1
3	40	40	40	165	624	0.1	2000	0.66
4	40	40	40	375	1040	0.1	2000	0.4
5	40	40	40	330	1248	0.1	2000	0.33

## Data Availability

Data are contained within the article.

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
