# Peer review of "Effect of Equivalence Ratio on Pollutant Formation in CH4O/H2/NH3 Blend Combustion"

_molecules, 2023, doi:10.3390/molecules29010176_

Round 1

Reviewer 1 Report

Comments and Suggestions for Authors

In this manuscript, molecular dynamics simulations of the reactive force field are utilized to investigate the pollutant formation characteristics of CH4O/H2/NH3 co-combustion and to analyze the mechanisms of CO, CO2, and NOx formation at different equivalence ratios at the molecular level. This study is important for controlling the generation and emission of C- containing pollutants and N-containing pollutants. This paper will be considered for acceptance after answering the following questions.

1. The discussion of free radicals in section 2.1 is not well developed.

2. Lack of error analysis of data changes in the text.

3. There are few instances where the language is ambiguous. Such as “This may be due to the fact that φ=0.5 is the highest CO peak. This may be due to the fact that there is more CO formation and less CO consumption at φ=0.5”.

4. The summary of conclusions is not concise enough, please simplify the presentation of conclusions

5. This article still has some formatting problems. Please modify it further.

Comments on the Quality of English Language

As above

Reviewer 2 Report

Comments and Suggestions for Authors

The effects of different equivalence ratios on combustion and emission characteristics are investigated of CH4O/H2/NH3 ternary fuels in this manuscript.

1. Line 64, “thin combustion” is better be changed as “fuel-lean combustion”

2. CH4O/H2/NH3 serves as green fuel and has been incurred many interests especially in recent years. But how can these fuel be supplied simultaneously in real devices? Especially at normal condition, methanol is liquid, ammonia and hydrogen are gas. Authors need to clarify this in the introduction.

3. Conclusion part needs to be summarized and highlights the main point. 

Comments on the Quality of English Language

Moderate editing of English language required

Reviewer 3 Report

Comments and Suggestions for Authors

The authors studied the combustion emissions of CH3OH/H2/NH3 blend which is a very promising fuel. I have the following comments and questions:

What is the pressure condition of your simulation?

How long does the calculation take when the simulation time is 1.25 ns? Do you expect a different conclusion when you have longer simulation time?

In Figure 2, why did the H2O molecules decrease with equivalence ratio? I suppose at equivalence ratio of 0.33 with higher numbers of OH and H, the numbers of H2O molecules should be higher than the case at equivalence ratio of 1.0.

In Figure 5, the authors may want to include and discuss more details of the coupling effects between NH3 and CH3OH, which is the most important part of blending fuel. Furthermore, I am confused why the major product of H-abstraction of CH3OH is CH3O? I suppose the major product is CH2OH due to the weaker bond energy of C-H than O-H in CH3OH.

Comments on the Quality of English Language

Minor editing of English language required

Round 2

Reviewer 2 Report

Comments and Suggestions for Authors

The mentioned comments have been well revised.

Author Response

谢谢你的建议。

Reviewer 3 Report

Comments and Suggestions for Authors

Accept

Author Response

谢谢你的建议。